# Topological Ensemble Detection with Differentiable Yoking

**David Klindt**                                              DAVID.KLINDT@GMAIL.COM
**Sigurd Gaukstad**                                      SIGURD.GAUKSTAD@NTNU.NO
**Melvin Vaupel**                                            MELVIN.VAUPEL@NTNU.NO
**Erik Hermansen**                                        ERIK.HERMANSENS@NTNU.NO
**Benjamin Dunn**                                          BENJAMIN.DUNN@NTNU.NO
*Department of Mathematics, NTNU*

**Editors:** Sophia Sanborn, Christian Shewmake, Simone Azeglio, Arianna Di Bernardo, Nina Miolane

## Abstract

Modern neural recordings comprise thousands of neurons recorded at millisecond precision. An important step in analyzing these recordings is to identify neural ensembles — subsets of neurons that represent a subsystem of specific functionality. A famous example in the mammalian brain is that of the grid cells, which separate into ensembles of different spatial resolution. Recent work demonstrated that recordings from individual ensembles exhibit the topological signature of a torus. This is obscured, however, in combined recordings from multiple ensembles. Inspired by this observation, we introduce a topological ensemble detection algorithm that is capable of unsupervised identification of neural ensembles based on their topological signatures. This identification is achieved by optimizing a loss function that captures the assumed topological signature of the ensemble. To our knowledge, this is the first method that leverages global features of the dataset to identify neural ensembles. This opens up exciting possibilities, e.g., searching for cell ensembles in prefrontal areas, which may represent cognitive maps on more conceptual spaces than grid cells.

**Keywords:** topological data analysis, ensemble detection, grid cells

## 1. Introduction and Background

The brain is classically described as being made up of brain areas of distinct functions such as the visual or motor cortices, each with neurons having response profiles, tuned to specific features of the animal's behavior. Grouped according to their selectivity, the neurons together encode internal representations of the physical variable. Thus, in cases where the encoded variable is recorded, one has historically used this additional information to cluster the neural data into populations of neurons (e.g. in Gardner et al. (2022)). This approach, however, appears to be overly simplistic when considering that, for instance, in visual areas there exists a substantial number of neurons with strong locomotive modulation (Keller et al., 2012) and still more of undetermined function (Olshausen and Field, 2005). A different approach is via the supposition that functional connectivity between neurons is exposed through the pairwise correlations of the activity Friston (1994), and that ensembles of neurons can be identified (clustered) as subgroups of neurons with high internal correlation Eggermont (2006). However, high correlations may be the result of many factors such as common global input or correlations between representations and not reflect the specific computation of the ensemble. Hence, the question becomes how to identify which neurons

belong to which group, especially when the tuning of the neurons is either unknown or unclear.

The medial entorhinal cortex is a brain area with neurons that have a variety of different response properties: grid cells Hafting et al. (2005), head direction Sargolini et al. (2006), border Solstad et al. (2008), speed Kropff et al. (2015) and object-vector cells (Høydal et al., 2019; Obenhaus et al., 2022). Grid cells are known to group in *ensembles* (i.e., modules), each ensemble tiling the space in which an animal navigates into hexagonal patterns of different spatial resolutions (Fyhn et al., 2007; Stensola et al., 2012). Recent work used persistent homology to demonstrate that the joint activity of grid cells from an individual ensemble resides on a toroidal manifold (Gardner et al., 2022) and that we may also think of each grid cell as having a receptive field, or tuning curve, on the surface of a 2-torus (Curto, 2017) (Appendix, Fig. 4). Crucially, the toroidal organization becomes apparent only after separating the distinct grid cell ensembles (see Rebuttal Fig. 4 in Peer Review File in Gardner et al. (2022)), i.e., it is a signature of the individual ensembles. Usually, the ensembles are separated based on their spatial autocorrelation functions (Gardner et al., 2022) which requires recording the animal's spatial position in an open field. Hence, in the absence of such additional information, or in tasks where the specific covariate is unknown (Constantinescu et al., 2016), we require an unsupervised clustering approach that infers the distinct cell ensembles. We propose doing this solely based on the topological structure of their population activity.

Topological data analysis (TDA) studies the topological shapes in datasets (Wasserman, 2018). A key instrument in the toolkit of TDA is the computation of persistent homology. This is visualized as a persistence diagram, where each point represents the introduction and depletion of an $n$-dimensional hole found in the dataset. The assumption is then that the long-lived holes are the most representative of the dataset (Ghrist, 2008). This has previously been used to discover the ring topology in head direction neural circuits (Rybakken et al., 2019), portraying a single long-lived homology class in dimension 1. We note that the statistics of barcodes are still actively studied (Curto et al., 2021) and which features are determined to be significant are often based on heuristics (Wasserman, 2018; Kang et al., 2021; Gardner et al., 2022). We propose an optimization objective which maximizes the lengths of a fixed number of significant persistent homology bars while minimizing all others. Optimizing in the discrete space of possible neural subsets requires a nontrivial smoothing step that we term 'differentiable yoking'. Essentially, we use stochastic gradient descent to learn a linear transformation of the data which converges to a repeated permutation of the correct ensemble. This approach squares the number of required parameters, which we argue makes the difficult combinatorial problem more easily solvable in an interpolative overparameterisation regime (Loog et al., 2020). As a *proof-of-concept* this is validated on simulated data.

## 2. Methods

### 2.1. Problem Setting

Let $X \in \mathbb{R}^{t \times n}$ be a recording of $t$ instantaneous firing rates from $n$ neurons. The Rips filtration of $X$ describes a sequence of approximated underlying spaces of $X$ across a given scale (in our case the pairwise Euclidean distances), connecting the points of $X$. Let $PH_k(X)$ de-

note the persistent homology of dimension $k$ of the Rips filtration, which can be decomposed into a sum of elementary persistence intervals (Carlsson, 2009)

$$PH_k(X) \simeq \bigoplus_i I([b_i, d_i]). \tag{1}$$

This describes the existence of $k$-dimensional holes in the filtration and $[b_i, d_i]$ will be referred to as the corresponding *bars*, depicting the scales at which a hole appears ($b_i$) and disappears ($d_i$), $d_i - b_i$, thus referred to as the *lifetime*.

We wish to find a subset $E \subseteq X$ with $E \in \mathbb{R}^{t \times m}$, such that the $m \leq n$ neurons form an *ensemble* with a toroidal state space. Specifically, our hypothesis is that $E = \{v_0, \ldots, v_t\}$ consists of $t$ (noisy) samples from the surface of a $p$-dimensional Torus $\mathbf{T}^p$. Note that here we focus on ensembles with state space activity dominated by a single $p$-dimensional torus, but the approach can be generalized to seek other spaces. However, we focus on $T^p$, as this may include several neural representations, e.g., head-direction cells (Taube et al., 1990), spinal motor networks (Lindén et al., 2022) and orientation-tuned cells (Hubel and Wiesel, 1959) are thought to be described as $T^1$; 2-D grid cells (Hafting et al., 2005) and 3-D head direction cells (Finkelstein et al., 2015) by $T^2$ and 3-D grid cells (Ginosar et al., 2021) and conjunctive grid-by-direction cells (Sargolini et al., 2006) as $T^3$.

## 2.2. Topological Loss Function

A torus is the product space of circles, i.e. $\mathbf{T}^p = S_1 \times \ldots \times S_p$. As each circle encloses a 1-dimensional hole, we would expect samples from a $p$-dimensional torus to give $p$ long-lived holes $J = \{j_1, \ldots j_p\}$ in dimension 1, while the remaining bars $[b_i, d_i]_{i \notin J}$ could be attributed to noise,

$$PH_1(E) \simeq \left( \bigoplus_{i \in J} I([b_i, d_i]) \right) \oplus \left( \bigoplus_{i \notin J} I([b_i, d_i]) \right). \tag{2}$$

We make it the objective of our ensemble detection method to maximize the lifetimes of the $p$ bars found in a subset point cloud $E$ and minimize those of the remaining bars. This will find the set of neurons whose activity strongly suggests a distinct toroidal representation, reflecting a pronounced functional relation between the neurons (Curto, 2017). Note that we only study the 1-dimensional holes in this paper, due to the computational complexity of computing higher-dimensional features. One such loss function would simply be

$$\mathcal{L}(X, W, p) := \sum_{i \notin J}(d_i - b_i) - \sum_{i \in J}(d_i - b_i). \tag{3}$$

Here $W$ is a binary weight matrix of the columns $X$, excluding the influence of one or more neurons, and $J$ identifies the $p$ longest-lived bars of $PH_1(XW)$. To find $E = (XW^*)_+$, where $W^* = \min_W(\mathcal{L}(X, W, p))$ and $(\ldots)_+$ gives the non-zero columns, we have to test among all possible subsets of $X$. Unfortunately, even if we know $m$ we still have $N = \binom{n}{m}$ distinct subsets to pick from. Thus, the brute force combinatorial approach does not scale beyond a few neurons.

### 2.3. Differentiable Yoking

Luckily, $\mathcal{L}$ is differentiable almost everywhere (Solomon et al., 2021; Gabrielsson et al., 2020) and we can approach the discrete search (i.e. clustering), for $E$ in the following way Bjerke et al. (2022). Let $W \in \mathbb{R}^{n \times n}$ be a real weight matrix, and

$$\sigma(W)_{:,j} := \frac{\exp W_{:,j}}{\sum_i \exp W_{ij}} \tag{4}$$

the softmax function applied to each column of $W$. We may then define a linear function, $f_W : \mathbb{R}^{t \times n} \to \mathbb{R}^{t \times n}$, multiplying the dataset, $X$, with the transformed weight matrix $\sigma(W)$, i.e. $f_W(X) := X\sigma(W)$. Importantly, each column $f_W(X)_{:,i} \in \mathbb{R}^n$ is then a *convex combination* of columns in $X$, since $\sum_i \sigma(W)_{ij} = 1, \forall j$ and $\sigma(W)_{ij} > 0, \forall i, j$. We call this smoothing of weights for *yoking* and next see how it may be applied in procuring a subset $E$ minimizing the loss function.

First, we note that a permutation of the $m$ columns of $E$ corresponds to a shuffling of the basis, and simplical homology is invariant under the ordering of the basis. Thus, also $\mathcal{L}$ is invariant to input permutations and we expect $W^*$ to be approximately equal to a permutation of $\sigma(W)$. Moreover, none of the other neurons should be selected, i.e., $\sigma(W)_i \approx 0$ for any $i$ corresponding to a column not in $E$.

To finally select the neurons in the ensemble based on the optimal $\sigma(W^*)$ found in minimizing $\mathcal{L}$, we require the included neurons to have summed weights above some threshold, $\theta > 0$, i.e. the proposed ensemble is defined as

$$\hat{E} := \left\{ X_i | \sum_i \sigma(W^*)_{:,i} > \theta \right\}. \tag{5}$$

For computations we choose $\theta = \text{median}(\max_i \sigma(W^*)_i)$.

### 2.4. Teddy algorithm

Topological Ensemble Detection with Differentiable Yoking (Teddy) is based on two critical components: i) differentiable persistent homology; ii) a continuous relaxation of discrete optimization in an overparameterized model. We use differentiable persistence (Gabrielsson et al., 2020) to optimize a gradient directly for a combination of neurons that fits the expected signature of a toroidal ensemble. Overparameterisation (Loog et al., 2020) is used as a way to circumvent the discrete optimization of finding a subset of neurons, by searching instead for a convex combination of our neurons that minimizes the topological loss function. This, unnecessarily, increases the number of parameters in our model from $O(N)$ to $O(N^2)$, where $N$ is the number of neurons. However, early experiments showed that this was crucial for getting the optimization scheme to work. We leave further exploration of this phenomenon for future research.

In the following we give the schematics of the algorithm. In short, we first randomly initialize the weight matrix $W$ and compute the persistent homology of $X\sigma(W)$. Only a small batch of $X$ is used in this computation to lessen the computational burden. $W$ is then updated through the gradient of the loss function and the procedure repeated until it reaches a minimum (or a given number of iterations).

---

**Algorithm 1:** Teddy

---

**Input:** $X \in \mathbb{R}^{t \times n}$ - Neural recording
**Parameter:** $p$ - Number of $H_1$'s, $\gamma$ - Learning rate, $\theta$ - Threshold for acceptance
**Output:** $E \in \mathbb{R}^{t \times m}$ - A submatrix of $X$

1 $E \leftarrow X$
2 $W \leftarrow \text{NormalMatrix}(0, 1)$
3 $W_{:,j} \leftarrow \text{Softmax}(W_{:,j})$
4 **while** $\mathcal{L}(E, W, p) \neq \min\{\mathcal{L}(E, W, p)\}$ **do**
5 $\quad$ $W \leftarrow W - \gamma \nabla \mathcal{L}(E, W, p)$
6 $\quad$ $W_{:,j} \leftarrow \text{Softmax}(W_{:,j})$
7 $\quad$ $E \leftarrow XW$
8 **end**
9 $E \leftarrow X[\max(W_{:,j}) > \theta]$

---

## 3. Experiments

### 3.1. Data Generation

We generate grid cell responses (on a 2-torus, i.e., $\mathbf{T}^2$) and, for comparison, neural ensembles with population activity living on other manifolds, including: $S^1$, $\mathbb{R}^2$ and SO(3). To this purpose, we proceed as follows: 1. Generate random points on the manifold as neural receptive field centers. 2. Generate random points on the manifold as latent/covariate states. 3. Compute the geodesic distances between the two sets of points. 4. Compute Gaussian bump-like tuning curves based on the geodesic distances. 5. Add Poisson noise (Appendix, Fig. 5).

Because each manifold has a different metric, we find that we need to adjust the width of the tuning curves and their relative response scaling to achieve homogeneously active populations of neurons for each of the manifolds. Thus, we optimize over tuning width and response scaling to find ensembles with a biologically plausible firing rate with mean and variance (same for Poisson) at approximately 5Hz.

### 3.2. Persistent Homology of Neural Responses

We use the 'VietorisRipsPersistence'-class from 'Giotto-TDA' to compute the persistent homology (up to dimension 2) for the neural response matrices from each latent manifold (Tauzin et al., 2021). Importantly, we treat each instance in time as a separate sample in the neural state space of population activity, which is of the same dimensionality as the number of neurons.

The persistence diagrams are as expected. For $\mathbf{T}^2$ we find two clear circles ($H_1$) and one prominent volume in dimension two ($H_2$). For $\mathbb{R}^2$ we find a few spurious $H_1$ and $H_2$, but no significant topological structure – however, these spurious features will become important below. For $S^1$ we find the signature of one long-lived circle ($H_1$). Finally, for SO(3) one would expect clear features in $H_0$ (which we always see) and $H_3$ (which we do not measure, due to the computational burden) (see Bubenik and Kim, 2007; Hatcher, 2005).

The crucial question is now: *What happens if we mix multiple ensembles?* Since we are mainly interested in grid cells, we will test what happens to the topological structure of neural population activity when a toroidal ensemble is paired with each of the other manifolds (including $\mathbf{T}^2$ itself to study the case of multiple mixed grid cell ensembles Stensola et al. (2012)). Let therefore $\mathbf{T}^2 \times X$, for $X \in \{\mathbf{T}^2, \mathbb{R}^2, S^1, \mathrm{SO}(3)\}$, denote the point cloud induced from the point clouds in the product of the ambient spaces. In almost all cases (see Fig. 1), pairing different ensembles hides the topological signature of $\mathbf{T}^2$ that we would have hoped to discover in the data. Interestingly, pairing neural ensembles from $\mathbf{T}^2 \times S^1$ does not completely abolish the strong top $H_1$ signature of $S^1$ in the persistence diagram.

### 3.3. Evaluating a Topological Loss Function for Separating Ensembles

We test whether the proposed topological loss function is able to tell apart toroidal ensembles (grid cells) when mixed with a varying number of neurons from other ensembles. The mixing degree is given in percentage of neurons from another ensemble relative to the toroidal ensemble. Furthermore, we assess the minimum number of temporal samples required to make a reliable inference, motivated by the fact that computing persistent ho-

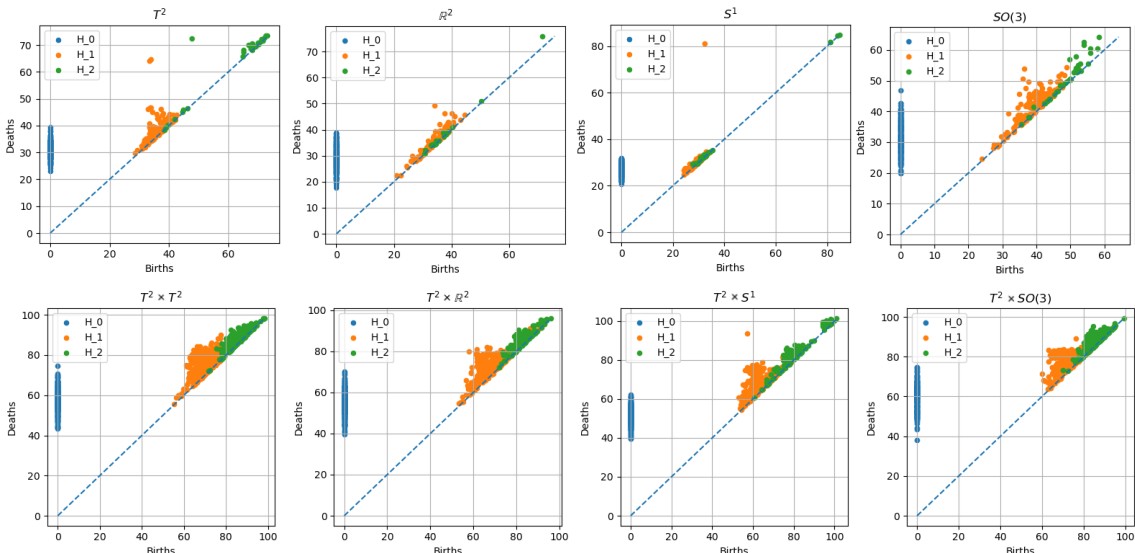

Figure 1: **Persistent Homology of Neural Responses**. Top, persistence diagrams of single neural ensemble responses with underlying manifold representations: $\mathbf{T}^2, \mathbb{R}^2, S^1, \mathrm{SO}(3)$. The diagrams show time of birth ($x$ axis) and death ($y$ axis) of zero- ($H_0$), one- ($H_1$) and two-dimensional ($H_2$) holes. Bottom, persistence diagrams of mixed neural population responses. Note the lack of clear topological signatures (long-lived holes), indicating these are obscured in joint recordings of distinct ensembles. We would however expect the homology of the combined state space to be found given more samples.

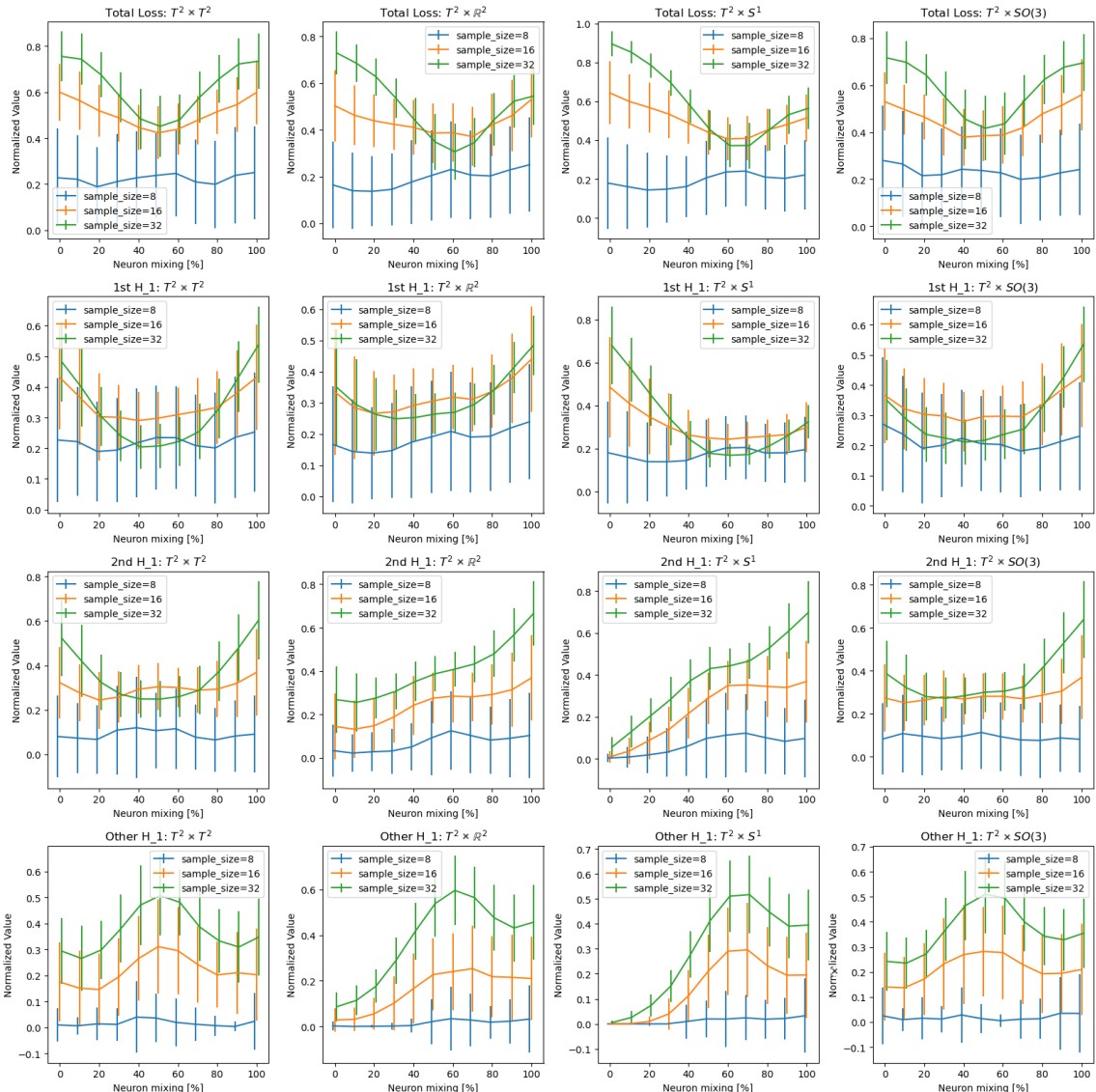

Figure 2: **Topological Loss Function**. The behaviour of the (features in the) loss function $\mathcal{L}(X)$ (eq. 3) was tested for a mixed population of neural ensembles from different manifolds: $X \in \{\mathbf{T}^2 \times \mathbf{T}^2, \mathbf{T}^2 \times \mathbb{R}^2, \mathbf{T}^2 \times S^1, \mathbf{T}^2 \times \mathrm{SO}(3)\}$. The size of $X$ was held fixed and the $x$-axis in each plot indicates the percentage of neurons coming from the $\mathbf{T}^2$ ensemble. High values on the right indicate a preference for selecting the $\mathbf{T}^2$ ensemble, whereas high values on the left indicate a preference for selecting the other ensemble in each comparison. Each row shows different measures of the persistence diagrams. The first row shows the normalized negative of the loss function. The second (third) row gives the lifetime of the (second) longest-lived $H_1$ in the persistence diagram. The fourth row gives the sum over all other $H_1$ lifetimes.

mology is computationally expensive for a large number of points Somasundaram et al. (2021). We list the following observations (Fig. 2):

- For $\mathbf{T}^2 \times \mathbf{T}^2$, as little as 16 temporal samples gives an increase in the topological objective function as we move from a perfectly mixed (mixing= 50%) to a perfectly separated (mixing= 100%) population of neurons. The lifetimes of both top $H_1$ circles increases in this process. Interestingly, in the sum of lifetimes over all other $H_1$ circles, we first observe a dip and then a slight increase for increasing mixing. This increase explains the levelling of the total loss. Moreover, as expected, the curve is symmetric in both directions since we have two symmetric toroidal ensembles.

- For $\mathbf{T}^2 \times \mathbb{R}^2$, there is a small increase in loss function with a larger percentage of $\mathbf{T}^2$-cells compared to perfectly mixed. However, the opposite gives an even higher increase. This is partially driven by a spurious $H_1$ (but no second $H_2$) appearing in the $\mathbb{R}^2$ ensemble. In addition, it is clear that we get increasingly more $H_1$ circles in the data as we include more neurons from the $\mathbf{T}^2$ ensemble. This is expected as a torus is made up of these. In composition this results in a total loss which overall favours the $\mathbb{R}^2$ ensemble.

- For $\mathbf{T}^2 \times S^1$, the analysis is similar to the one for $\mathbf{T}^2 \times \mathbb{R}^2$. However, we now know that the top $H_1$ is significant reflecting a true feature of the $S^1$ ensemble; and the second longest $H_1$ is even more clearly absent.

- For $\mathbf{T}^2 \times \mathrm{SO}(3)$, we see almost a symmetric total loss function. This results from a preference for the significant top two $H_1$ circles in $\mathbf{T}^2$ which are counteracted by the larger total number of circles in $\mathbf{T}^2$ (although we see also more circles in $\mathrm{SO}(3)$ than, e.g., in $S^1$ or $\mathbb{R}^2$ – which is expected).

### 3.4. Using Teddy to Separate Ensembles

We test the clustering performance of Teddy by assessing how many neurons it correctly identified as belonging to either of the two mixed ensembles tested. In these analyses we take random samples from a larger pool than what used for persistent homology (PH) computation above. As we are now looking for a distinct topological feature in the data, we will refer to the optimization of an objective function – the negative of the loss function. We see in Fig. 7 (Appendix) that Teddy is able to separate the two ensembles in all settings. Moreover, we make the following observations:

- For $\mathbf{T}^2 \times \mathbf{T}^2$, Teddy randomly (we check this below) selects one of the two tori. This requires *symmetry-breaking*, which we discuss below.

- For $\mathbf{T}^2 \times \mathbb{R}^2$, Teddy selects $\mathbb{R}^2$. We saw above that $\mathbb{R}^2$ does indeed lead to a higher value in the topological objective function due to a combination of a spurious top $H_1$ and more non-significant $H_1$ circles in $\mathbf{T}^2$.

- For $\mathbf{T}^2 \times S^1$, Teddy selects $S^1$. Again, this is in line with our observations above where the top $H_1$ from $S^1$ clearly led to a higher value in the topological objective function (combined with fewer other $H_1$s).

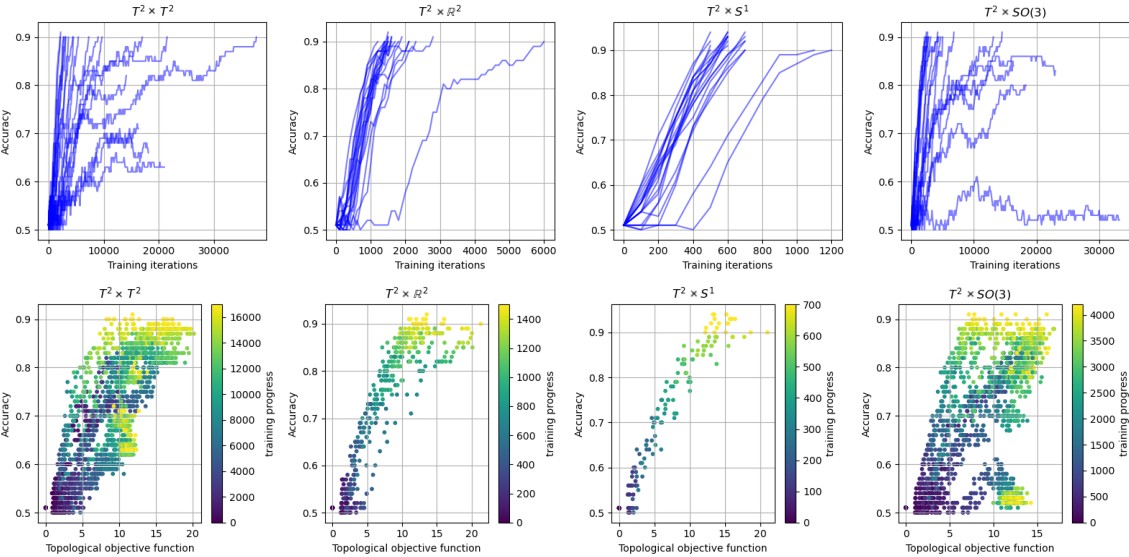

Figure 3: **Teddy Ensemble Detection Performance**. Top, $n = 20$ different runs (seeds) showing the ensemble detection accuracy of Teddy over the course of training (halted at 0.9 accuracy or stopping criterion to reduce computation time) for all four different mixed populations (columns). Bottom, ensemble detection accuracy for each run and time point as a function of the value of the topological objective function (negative of loss function in eq. 3). Colouring shows training progress, indicated by colorbar. The accuracy is positively correlated with the optimization target.

- For $\mathbf{T}^2 \times \mathrm{SO}(3)$, Teddy selects $\mathrm{SO}(3)$. This is somewhat surprising since we speculated above that both $\mathbf{T}^2$ and $\mathrm{SO}(3)$ present similar solution qualities in terms of the topological objective function.

Next, we further investigate how often Teddy prefers one solution over the other. To do this, we re-run each setting over a number of random seeds for data generation and initialisation. This also allows us to study the average performance of our method (independent of the random seed in the simulations above). We get perfect results for $\mathbf{T}^2 \times \mathbb{R}^2$ and $\mathbf{T}^2 \times S^1$ with a success rate of 1 (see Fig. 3). For the other two settings $\mathbf{T}^2 \times \mathbf{T}^2$ and $\mathbf{T}^2 \times \mathrm{SO}(3)$, we get a decent success rate of 0.8. Looking at the individual runs (plots in top row), we see that most runs seem to be going in the right direction, but trigger the stopping criterion of our training. This could be fixed, but would increase the run time. Judging by the average number of steps, we have our settings in increasing order of difficulty as follows: $\mathbf{T}^2 \times S^1$, $\mathbf{T}^2 \times \mathbb{R}^2$, $\mathbf{T}^2 \times \mathrm{SO}(3)$ and $\mathbf{T}^2 \times \mathbf{T}^2$. Furthermore, we note that the setting $\mathbf{T}^2 \times \mathbb{R}^2$ is likely biased towards picking $T^2$ as expected (with $\mathbb{R}^2$ not containing any 1-D holes). However, in all other settings Teddy is clearly biased towards picking the alternative ensemble. While this solves the task we set out to do, it still provides food for thought and further study.

In the bottom row of Fig. 7 (Appendix), we see that the value of the topological objective function is strongly related to the accuracy of our learning algorithm. That is, larger values in the objective are positively correlated with better task performance, which shows the consistency of our approach. This also suggests a simple remedy for the runs that did not converge: run the algorithm over a few random seeds and pick the solution with the highest objective function value. Ideally, that should recover the right solution even if individual runs got stuck in local minima. A similar approach is the default setting for sklearn's implementation of KMeans (see the 'n_init' parameter there).

## 4. Conclusion

In this work we have provided a *proof-of-concept* for two data science challenges that we find exciting. Firstly, we derived an unsupervised clustering algorithm with a topological loss function that can separate simulated neural ensembles tuned to distinct latent variable manifolds, including $\mathbf{T}^2, S^1, \mathbb{R}^2$ and SO(3). Secondly, we proposed an overparameterized relaxation to a discrete optimization problem which we believe merits more research in the future (Loog et al., 2020). Specifically, we would like to get a better understanding of the consequences of computing a convex combination of the original data and its implications for the downstream computation of persistent homology.

Next, we have seen that our algorithm tends to separate ensembles, however, preferring the unintended solutions. Practically, we note that this is not a concern since we just need to separate the two ensembles (or more, provided repeated application of Teddy). We can then post-hoc always compute each ensembles' persistence diagram and choose the ensemble with the designated, in this case toroidal, signature. One might also wonder if we could set up our loss function differently, e.g., by also maximizing the lifetime of the top $H_2$ – but this increases the computation burden considerably. Another approach would be to only maximize the lifetime of the second longest $H_1$ or dropping the term in the loss function that aims at minimizing the sum over all other $H_1$. However, simulations (not shown here) have shown that, in the former case, while this more often selects the correct toroidal ensemble, it takes much longer to optimize because, effectively, we are only computing gradients for a single simplicial complex and thus much less of the data. In the latter case, simulations slows down substantially and also reduces the success of our optimization, possibly due to the convex combination of the original data that we are feeding into the persistence computation. We believe that further theoretical investigation will likely provide a more selective objective function that still results in a fast and robust optimization.

We have hinted at the problem of *symmetry-breaking* which comes up in the case of a mixed population where each ensemble comes from the same type of latent manifold ($\mathbf{T}^2 \times \mathbf{T}^2$ in our simulations). We note (results not shown) that changing the mean firing rate and, therefore, the signal-to-noise ratio (SNR) in one of the two ensembles immediately achieved this goal. Thus, this problem may be less of a nuisance in realistic settings where it is unlikely that we will ever record from two neural populations with identical SNR. Another limitation of our approach is that computing persistent homology is computationally burdening and also prone to noise. Further research is needed to investigate how both shortcomings might be improved (see Methods in Gardner et al. (2022)). One exciting direction might be a differentiable estimation of persistence metrics with neural networks (de Surrel et al., 2022).

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

# Appendix A. Appendix

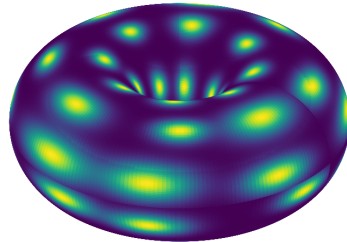

Figure 4: **Receptive Fields on Torus**. Illustration of the idea that each grid cell has a receptive field on a torus (Curto, 2017; Gardner et al., 2022). Each of the shown receptive fields (separated, bright spots) represent the selectivity of a single neuron. Thus, if the current toroidal representation location of the animal's location is close to the center of a neuron's receptive field, then this will elicit a strong response.

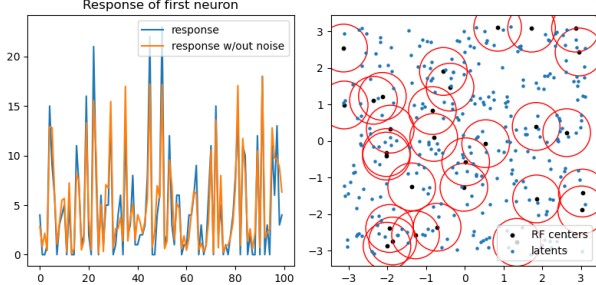

Figure 5: **Simulated Grid Cell Responses**. Left, example neuron's response (clean and noisy) to a 100 time steps of toroidal positions. Right, torus represented as a square $[-\pi, \pi]^2$ with circular boundaries (top folds to bottom, sides fold onto each other). Time points (representation of space in toroidal coordinates) are marked in blue and center/extent of reach neuron's receptive field (RF) in black/red.

## A.1. Baseline: KMeans Clustering

Here we assess how well neural ensembles can be separated with the classical KMeans algorithm, which is loosely inspired by previous work (Hamm et al., 2021; Lopes-dos Santos et al., 2011; Baden et al., 2016). Being a critical case (with an eye on grid cell research), we test this on the pairing of two neural ensembles each tuned to covariates on $\mathbf{T}^2$. We perform KMeans in the space of principal directions of the data and assess the clustering performance (with known ground truth) as a function of the number of directions included.

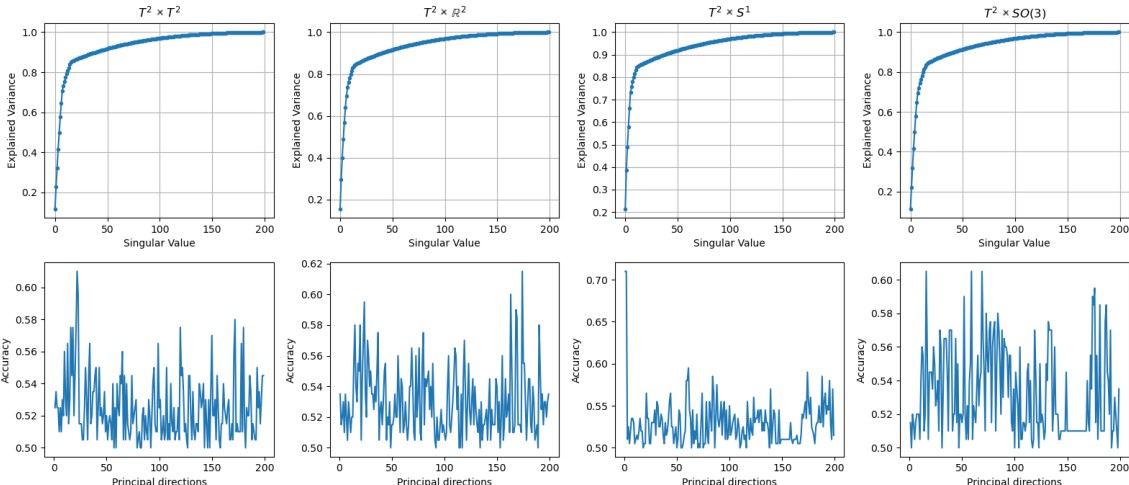

Figure 6: **KMeans Ensemble Detection**. Top, spectrum, i.e., cumulative variance explained by singular values of neural population activity for the distinct mixtures. Bottom, ensemble separation accuracy as a function of the number of principal directions included into a standard KMeans clustering of the resulting low-dimensional projection of the population activity.

Chance level performance is at 50%. However, since both ensembles are identical we consider each possible labelling of the two ensembles equally valid. Nevertheless, we find that for all dimensions of the data, KMeans is not able to separate the two toroidal ensembles with an accuracy above 62% (Appendix, Fig. 6).

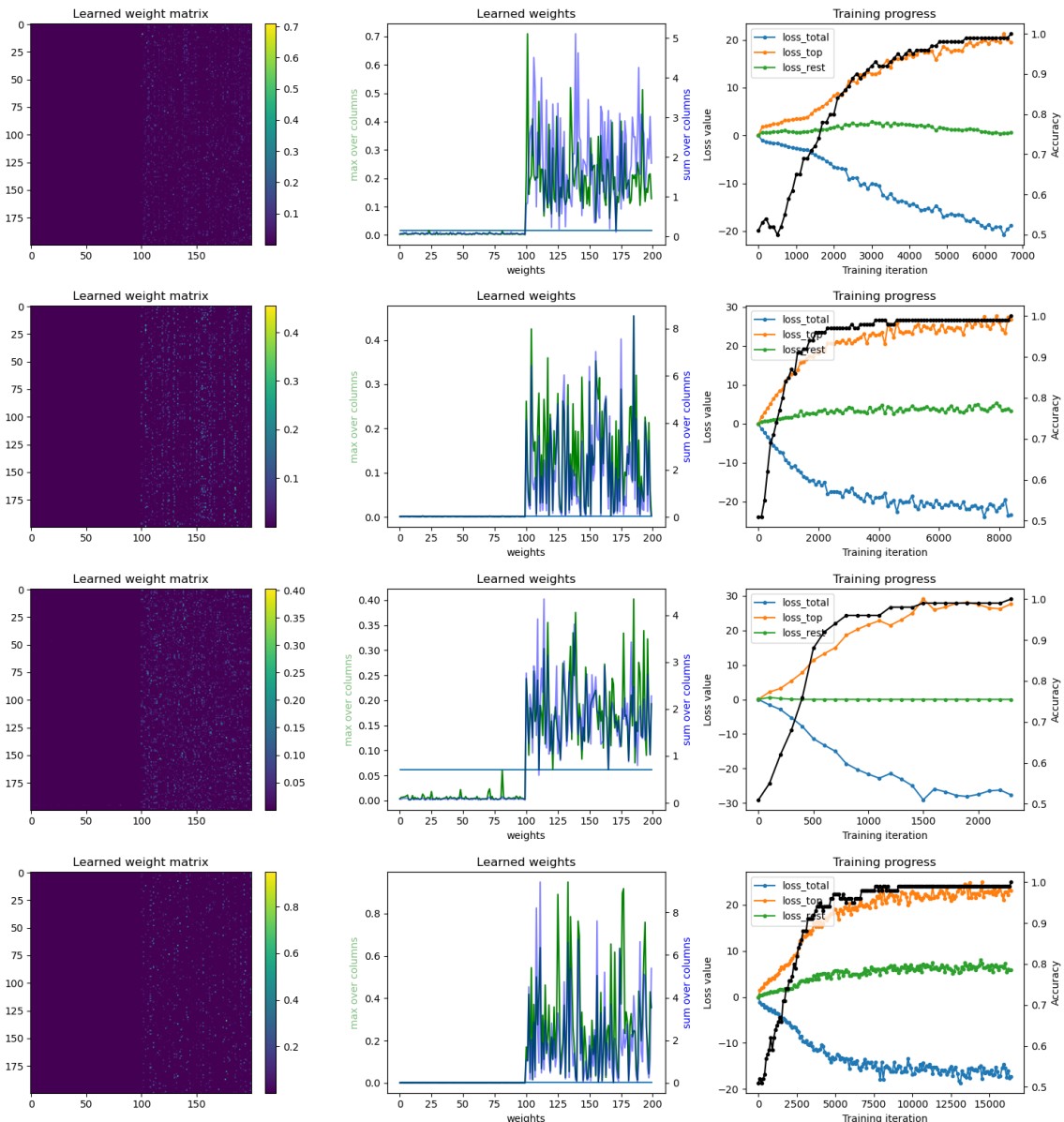

Figure 7: **Example Optimization Run**. The four rows show Teddy optimization for all settings (in order: $[\mathbf{T}^2 \times \mathbf{T}^2], [\mathbf{T}^2 \times \mathbb{R}^2], [\mathbf{T}^2 \times S^1], [\mathbf{T}^2 \times \mathrm{SO}(3)]$). Left, shows the final trained weight matrix, clearly placing higher values (see colorbar) on the left half of the column belonging to the same ensemble. Middle, maximum (green/left) and sum (blue/right) over the rows of the learned weight matrix. Right, loss functions over time: total loss (blue), sum over two longest $H_1$s (orange), sum over remaining $H_1$s (green) and ensemble detection accuracy (black, left).

