# OpenReview forum: "Topological Ensemble Detection with Differentiable Yoking"
_NeurIPS.cc/2022/Workshop/NeurReps — NeurReps 2022 Poster_

### Official Review · Reviewer_cgh8 · 2022-10-15
**Review of Topological Ensemble Detection with Differentiable Yoking**

**Confidence:** 4
**Soundness:** 2
**Presentation:** 2
**Contribution:** 1
**Overall Rating:** 3

**Summary:**

This paper considers the problem of ensemble identification --- subsets of neurons that represent a subsystem of functionality. The paper considers the specific case of grid cells in the mammalian brain, which are well-modeled by a torus, as a specific example. The method TEDDY, Topological Ensemble Detection with Differentiable Yoking is a way to perform unsupervised identification of neuronal ensembles based on their topological signatures. The method proceeds by optimizing a loss function encouraging a presumed topological signature.

**Questions:**

What is your mathematical definition or model for an ensemble?

What is the main idea for why minimizing a loss function defined in terms of persistent homology allows you to identify ensembles?

What is the definition of T^2 + T^2, or S^1 + T^2, etc? In my opinion this is never made clear.

**Limitations:**

The basic concepts being studied (ensembles) are not properly defined mathematically, or explained how they are being modeled mathematically. The problem statement is unclear. Advanced methods (gradient descent) is being performed, but it is not clear why these advanced methods are helpful for the problem of interest.

**Recommended Decision:**

1: Reject

**Relevance:**

2: Limited relevance

**Strengths And Weaknesses:**

A strength of the paper is that it leverages creative techniques to address a difficult problem. A weakness of the paper is that the translation into mathematical terms is not fully well-defined or well-explained. An ensemble is defined in the second sentence of the abstract, as a subset of neurons that represent a subsystem of specific functionality. However, in Section 2.2, which is mathematical, an ensemble is not defined. In one place, the authors write E={E_0,...E_t} but it is not clear the what the different sets E_i are. In (4), the authors write E = min ..., i.e. they give a different expression entirely. It is not clear how a subset of neurons that represent a subsystem of specific functionality is being translated into mathematical terms. Mathematically, is this modeled as a set of points that are nearby in some space? Or as something else? A second weakness of the paper is that they rely too much on a torus assumption. Prior work shows that the grid cells in a mammalian brain can be modeled as a 2-dimensional torus. Why do the authors then consider a p-dimensional torus? If the method only applies when one has a pre-assumed topological space model for the neural activity in mind, then this is a limitation (in general the underlying space modeling neural activity is very mysterious).

**Submission Track:**

Proceedings Paper (9 Page)

---

### Official Review · Reviewer_xFd3 · 2022-10-15
**Clustering in a generative model with a strong topological prior is an interesting problem**

**Confidence:** 3
**Soundness:** 2
**Presentation:** 2
**Contribution:** 1
**Overall Rating:** 3

**Summary:**

Clustering of synthetic 3-dim point clouds sampled from a geometrically strongly underspecified mixture model over primitive topological structures is suggested as a model problem for the task of detecting neural ensembles, and is approached computationally by gradient-based optimisation, of relaxed assignment weights whose number is the square of the number of inputs, for a loss function heuristically defined through persistent homology.

**Questions:**

- Is L a function of I rather than X in eq. (3)? Presumably, the optimisation is
  over cluster assignments rather than over data? Overall, the notation in eq.
  (3-6) is rather confusing, and would benefit from using standard notation from
  the clustering literature.

- Please use a proper generative model in statistical notation to make your experiments interpretable and reproducible. In particular:
  - What is the geometric arrangement of the manifolds which are "paired with
    each other", i.e., their sizes, orientations and distances? What is the
    distribution used for sampling from "R^2", and which plane is that actually
    denoting within R^3, which is presumably your ambient space? I would expect
    very different topological signatures depending on those choices. The
    results are essentially non-reproducible without this information.
  - What is "degree of mixing"? The caption of Fig. 2 suggests that it might be
    the mixture weight of the T^2 component of your mixture model, but otherwise
    this is a rather confusing term.

- Please include a proper algorithm block to make your experiments interpretable
  and reproducible.

**Limitations:**

-

**Recommended Decision:**

1: Reject

**Relevance:**

3: Solid fit

**Strengths And Weaknesses:**

-

**Submission Track:**

Proceedings Paper (9 Page)

---

### Official Review · Reviewer_QQ9C · 2022-10-19
**Teddy - a method to detect multiple neural ensembles that encode distinct topologies**

**Confidence:** 3
**Soundness:** 3
**Presentation:** 3
**Contribution:** 3
**Overall Rating:** 8

**Summary:**

The authors tackle the problem of different subpopulations of neurons coding for different topological classes, and the combinatorial complexity associated with separating and identifying them. Specifically, a recent paper identifies different grid cell modules code for different tori, which can each be identified by their persistent homologies if the module identity is known. The authors choose to formulate a method without knowing such external covariates.

They form a loss function that can maximize the life of significant bars within the persistent homology of the data, thereby picking a limited number of ensembles representing a particular topology. The loss function is shown to be differentiable, allowing the use of gradient descent and clustering to separate the ensembles. They examine the evolution of this loss function for different sample numbers and different mixing ratios of neurons between one manifold (T2) and four others (T2, R2, S1 and SO(3)) and determine that with a small number of samples (16-32) the loss function can determine one of the ensembles. This means that repeated runs of the algorithm may be able to separate multiple mixed ensembles.

The method (Teddy) uses the differentiable loss function with overparameterization to find the convex combination of neurons that minimizes the loss function (maximizes the objective function). They show almost perfect results with T2+R2 and T2+S1, and good but not perfect results with T2+T2 (due to symmetry) and T2+SO(3) (due to shared PH bars). The value of the objective function is related to the accuracy of the algorithm, indicating its importance in the success of the method.

**Questions:**

Specific comments:

Introduction - I disagree that representations need to be of physical variables. While sensory and motor correlates are certainly found in neural circuits, several brain regions code for abstract cognitive variables, driven by internal dynamics. It is precisely because of this that we want to use TDA independent of external covariates.

Eq (2) The variable I seems to be overloaded here - it is used to represent the interval [b_i, d_i] and also refer to the set {j_1...j_p}.

Page 5: Please refer to Fig. 1 here - it is currently not referenced anywhere

Page 5: "we find two significant circles" - the term significant implies a statistical test, and I do not see that one was performed. Please use a different term.

Figure 1: For those not familiar with persistence diagrams, what constitutes a significant point?  It would be good to say this in the caption or indicate in the figure.

Accuracy of clustering is the primary metric that is used in the paper, but it is not defined anywhere. It may be pretty trivial but it is worth defining at least in text

Fig. 2: Please show y axis as "Topological objective function" and not "Normalized value".

Fig. 2: "x-axis in each plot indicates how many neurons came from the T2 ensemble".  Though this is technically correct, the x axis is indicated as neuron mixing percentage. Please revise the caption to clarify.

Fig. 2 Would be helpful to put an arrow on the x-axis indicating the T2 direction (right) and the 'other' direction (left)

To improve clarity, In all figures (esp. Fig. 1 and 2), the authors are advised to increase the font size and increase the size of the subpanels where there is too much white space. It is good practice to improve the clarity and not directly use the figures that are created by the analysis program

Using both loss function and objective function in the manuscript is confusing to the reader, especially since in some places (like Fig. 1) it is not labeled. I would suggest using just one of the terms and talk about e.g. minimizing the loss function as compared to maximizing the objective function.

What is the mixing ratio for the analysis in Fig. 2? I was not able to find it anywhere.

Page 8: "we note that the settingT2 + R2 is likely unbiased (chance level being bias=0.5) as expected." What does this mean?

Fig. 5: In the right plot, please indicate the neuron center / extent whose response is being shown on the left plot.

Fig. 5: The term 'time points' seems to indicate some underlying continuous trajectory that you are sampling from. Since this is not the case, I would suggest calling these 'samples' instead.

Fig. 6: It is unclear what the takeaway of this figure is. I would suggest either explaining more in the caption or omitting this figure entirely.

Fig. 7. The left column figures (weight matrices) are not clear. I suggest using a different color scheme - perhaps grayscale with white for 0? This way you will be able to clearly see the effect that you are trying to convey.



**Limitations:**

Suggestions are included in the previous section.

**Recommended Decision:**

3: Accept

**Relevance:**

4: Highly relevant

**Strengths And Weaknesses:**

Strengths:

Originality: The paper outlines a technique to separate subpopulations of ensembles of neurons which can encode multiple topologies. This is an important problem in the analysis of neural ensembles. Using the loss function and the overparameterization-based discrete optimization technique that the authors described allows separation between multiple distinct topological ensembles with impressive accuracy.

Quality: I believe the claims and the supporting analyses.

Clarity: The submission is well-organized, although the technique could be explained a lot better, as I am noting below.

Significance: The authors clearly explain the significance, novelty and application of their method. I find this dimension to be excellent.

Weaknesses:

1. I found Sec 2.3 to be very confusing - the loss function is shown to be differentiable, function is found that creates convex combinations of the data, and the loss function is found to be invariant to input permutations. What is not clear to me is how you are using the loss function in order to compute the correct set of permutations. There might be an obvious step here that you have excluded because field-specific knowledge is assumed, but it would be better if this section were revised for clarity.

2. Similarly, I found the explanation of the method itself to be somewhat lacking - you mention overparameterization and discrete optimization using the convex combination function, but I am left to put the pieces together as to how you are doing it. Again, there is a large amount of field-specific knowledge assumed for readers of this interdisciplinary workshop.

3. Although the authors examine the effect of mixing for their loss function, mixing ratios are not evaluated in their examination of the accuracy of their method. I believe this is a missed opportunity since we cannot control the mixing ratios in neural recordings. Showing that the method is robust to this uncontrolled factor is an important step to ensure its viability.

4. Although the authors discuss that three or more ensembles could be separated using this technique, an example of this as a supplementary figure would have significantly strengthened the paper.

**Submission Track:**

Proceedings Paper (9 Page)

---

### Decision · Program_Chairs · 2022-10-22

**Decision:**

Accept (Poster)

**Comment:**

Although the paper received a low average score (4.67), there is high variance in the scores (3, 3, 8). We note that one of the reviews (xFd3, overall rating 3) is incomplete.

Reviewer QQ9C notes that the authors tackle "an important problem in the analysis of neural ensembles" and propose an approach that "allows separation between multiple distinct topological ensembles with impressive accuracy."

Reviewer QQ9C also notes that the explanation of the method is confusing, which is reiterated by the other two reviewers cgh8 and xFd3.

Reviewer cgh8 finds the translation of the neuroscience problem into mathematical terms to be lacking. We agree that the paper would benefit from more time spent setting up the problem and explaining the neuroscience concepts and their formalization more clearly.

Despite these drawbacks, we assess the paper to be highly relevant both to the topic of the workshop and to the computational neuroscience community more generally. Moreover, we find the thorough demonstrations and analyses on synthetic data convincing enough to warrant inclusion in the workshop proceedings. We recommend that the authors address the critiques regarding clarity in the camera-ready submission.